# Evaluating the Quality of Latent Tuberculosis Infection Screening in Ireland: A Single-Centre Retrospective Cohort Study

**DOI:** 10.3390/tropicalmed7020019

**Published:** 2022-01-28

**Authors:** James O’Connell, Joy Oguntuase, Brian Li, Cora McNally, Debbi Stanistreet, Samuel McConkey, Eoghan de Barra

**Affiliations:** 1Department of International Health and Tropical Medicine, Royal College of Surgeons in Ireland, D02 DH60 Dublin, Ireland; coramcnally@beaumont.ie (C.M.); smmconkey@rcsi.ie (S.M.); edebarra@rcsi.ie (E.d.B.); 2School of Postgraduate Studies, Royal College of Surgeons in Ireland, D02 YN77 Dublin, Ireland; joyoguntuase@rcsi.ie; 3School of Medicine, Royal College of Surgeons in Ireland, D09 YD60 Dublin, Ireland; brianli@rcsi.ie; 4Department of Infectious Diseases, Beaumont Hospital, D09 V2NO Dublin, Ireland; 5Department of Epidemiology and Public Health Medicine, Royal College of Surgeons in Ireland, D02 DH60 Dublin, Ireland; debbistanistreet@rcsi.ie

**Keywords:** latent tuberculosis infection, quality of care, effectiveness, equity, cost analysis

## Abstract

Ireland is a country with a low incidence of tuberculosis (TB) (5.6 cases per 100,000 population in 2019) that should be aiming for TB elimination (fewer than 1 case per million of population). To achieve TB elimination in low-incidence countries, programmatic latent tuberculosis infection (LTBI) management is important. This requires high-quality latent tuberculosis infection (LTBI) screening. Aim: To assess the quality of LTBI screening in a tertiary centre in Ireland using a framework. Methods: A retrospective review of the health care records of patients screened for TB in a tertiary centre in Ireland using an interferon-gamma release assay (IGRA) between 2016 and 2018 was performed. Three domains from the Institute of Medicine framework for health care quality, effectiveness, efficiency, and equity, were applied to measure the quality of LTBI screening. Results: Forty patients had LTBI and an indication for treatment, of whom 20% (8/40) were not offered treatment by the health care provider, 2.5% (1/40) did not accept treatment, and 10% (4/40) did not complete treatment. Seventy-five percent (6/8) of patients not offered treatment were non-Irish. The cost of screening per LTBI case identified was EUR 2048. Conclusions: This study evaluated the quality of LTBI screening using a framework and identified that LTBI screening in this tertiary centre needs to be scaled and expanded, and that treatment initiation needs to be improved, particularly among non-Irish nationals.

## 1. Introduction

Tuberculosis (TB) is a leading cause of global morbidity and mortality, despite being preventable [1]. Ending TB is a global priority, as evident by the existence of the World Health Organisation (WHO) End Tuberculosis Strategy [2], which includes a target of a 90% reduction in TB incidence by 2035 (compared with 2015). As well as collaborating globally to achieve this target, the strategy requires all countries to adapt it at a national level. In 2020, there were 57 countries with a low incidence of TB (defined as fewer than 10 cases per 100,000) [1] that, as well as aiming to achieve the End TB Strategy targets, should be aiming to eliminate TB (defined as fewer than 1 case per million) [3], an important step in global aspirations to end the disease. TB in low-incidence countries is characterised by most disease (70–80%) being due to TB reactivation rather than recent transmission [4,5]. Therefore, identifying latent tuberculosis infection (LTBI) among population groups with a high prevalence or risk of reactivation and providing them with preventive care is important to achieve TB elimination in low-incidence countries [3]. Preventive care involves reducing risk factors for TB reactivation, such as smoking or excessive alcohol intake [6], but also providing chemoprophylactic treatment, which is highly effective [7,8].

Ireland is a high-income European country with a low incidence of TB (5.6 cases per 100,000 in 2019), drug-resistant TB, and human immunodeficiency virus (HIV)/TB coinfection [9]. The incidence of TB is declining, and deaths due to TB are infrequent [9]. Despite being well placed to eliminate TB, Ireland is not on target to do so [3]. The European Centre for Disease Prevention and Control recommend that low-incidence European countries, such as Ireland, programmatically screen and treat those at-risk of TB for LTBI [6]. At a minimum, this includes people living with HIV; immunosuppressed persons (such as patients on anti-tumour necrosis factor alpha treatment); patients preparing for transplantation; patients with end-stage renal diseases or preparing for dialysis; patients with silicosis; people with pulmonary fibrotic lesions; and contacts of infectious TB cases (based on a risk assessment of their exposure) [3,6]. In Ireland, TB care is primarily provided by specialists in tertiary centres and regional public health departments. There is no national programmatic approach to LTBI.

If the programmatic management of LTBI is to be an effective approach to eliminating TB in low-incidence countries, then LTBI screening must be of high quality to maximise those at-risk of TB identified for screening who would subsequently benefit from completing preventive treatment [10]. Therefore, evaluations that define, measure, and report screening quality are important. Such evaluations should be performed nationally to be informative for programmatic LTBI management, but also locally to be informative for local service provision [10]. Where deficits in screening quality are identified, health care providers can engage with their patients to understand why, and service quality can be improved. In Ireland, evaluations of LTBI screening are lacking across all at-risk groups, even at a local level [11]. There is a need for LTBI screening evaluations to be performed at service and national levels in Ireland. Without these, or any progress in establishing programmatic LTBI care, TB elimination efforts in Ireland will remain off track. This study aimed to evaluate the quality of LTBI screening in a tertiary centre in Ireland using a framework.

### Framework to Measure the Quality of Latent Tuberculosis Infection Screening

To evaluate quality, it is necessary to apply a framework that defines quality and outlines how it can be measured. A widely used framework for assessing quality is that of the Institute of Medicine [12], which has been adapted by the WHO and used in their framework for quality, as described in Delivering quality health services: a global imperative for universal health [13]. According to the Institute of Medicine, quality of care is defined as “the degree to which health services for individuals and populations increase the likelihood of desired health outcomes and are consistent with current professional knowledge” [12] and can be measured according to six domains; effectiveness, efficiency, equity, timeliness, patient-centredness, and safety (integration was also added to the Institute of Medicine framework by the WHO in their adaptation). 

## 2. Methods

### 2.1. Research Design, Setting, and Sample

A retrospective review of the health care records of all patients who underwent interferon-gamma release assay testing (IGRA) between 2016 and 2018 in a tertiary centre (Beaumont Hospital, Dublin, Ireland) in Ireland was performed. Figure 1 demonstrates the framework, its domains, and the measures applied in this evaluation. Three of the seven WHO domains were applied in this evaluation (effectiveness, equity, and efficiency). Effectiveness, which means providing care processes and achieving outcomes as supported by scientific evidence [12], was measured as the proportion of patients completing LTBI screening and who subsequently completed treatment. These are commonly used measures when evaluating LTBI services [14] and are used internationally in LTBI screening programmes [15]. Equity, which means providing health care of equal quality to those who may differ in personal characteristics other than their clinical condition or preferences for care [12], was assessed by comparing the outcomes of LTBI screening and treatment according to age, sex, and nationality. Efficiency refers to maximising the quality of a comparable unit of health care delivered or unit of health benefit achieved for a given unit of health care resources used [12]. The costs of screening per patient with LTBI identified and per patient with LTBI treated were estimated as a measure of efficiency. The cost per IGRA processed was EUR 50 [16]. The purchase cost of the test tubes was EUR 12.18 per set [17]. The cost of the phlebotomist obtaining the sample was calculated as EUR 8.49, using guidance from the Department of Public Expenditure and Reform [18]. The cost per IGRA test performed was EUR 70.67 (EUR 50 plus EUR 12.18 plus EUR 8.49). This study was registered as a clinical audit with the Beaumont Hospital Office of Clinical Audit (audit number 572). The Strengthening the Reporting of Observational Studies in Epidemiology (STROBE) checklist for cohort studies was used to report the findings of this study (Appendix A) [19].

### 2.2. Data Sources

The patients’ name, date of birth, date of test request, indication for testing, and test results of all IGRA samples sent to the laboratory directorate between 1 January 2016 and 31 December 2018 (inclusive) were provided by the laboratory directorate, who record these routinely in a Microsoft Excel file for all samples received. All other data collected in this evaluation not contained in the file received from the laboratory directorate were collected from patient health care records. Health care records consisted of emergency department notes, discharge summaries, outpatient clinic letters, radiology files, and the patient’s medical record file.

### 2.3. Data Collection

Data collection was performed by J. O’Connell, B. Li, and J. Oguntuase, using a data collection form on Microsoft Excel. All data collected were reviewed by J. O’Connell for accuracy. Where there was any disagreement about the accuracy of the data collected, the health care record was reviewed jointly by J. O’Connell and the other data collector, and mutual agreement was reached as to what the correct data point was. 

Screening completion was evaluated in a random selection of patients, representing approximately 10% of patients in the dataset, and in all patients with a positive IGRA. Screening completion required a diagnosis of active TB to be confirmed or excluded according to two criteria. The first criterion was that there was documentation at the time of IGRA request that the patient had no symptoms of TB, or if symptoms of TB (fever, fatigue, weight loss, night sweats, cough or shortness of breath or swollen glands (lymphadenopathy)) were documented, documentation of clinical or microbiological confirmation or exclusion of a diagnosis of TB, or documentation of an alternative diagnosis explaining the patient’s symptoms were evident in the patient’s health care record. The second criterion was that the patient had had a chest radiograph within two years of the IGRA request date and if there were any radiographic findings of active TB (consolidation, cavitations, pleural effusions, or hilar lymphadenopathy), documentation of clinical or microbiological confirmation or exclusion of a diagnosis of TB, or documentation of an alternative diagnosis explaining the radiological finding were evident in the patient’s health care record. 

After evaluating screening completion, the outcome of screening (active TB, LTBI, or no TB infection) for all patients within the dataset was evaluated. Patients with a clinical or microbiological diagnosis of TB documented in their health care record were categorised as having had TB. Patients with a diagnosis of LTBI documented in their health care record were categorised as having had LTBI. Patients with a positive IGRA and having had TB excluded (as described above) and no history of previously treated TB or LTBI were categorised as having had LTBI. 

For patients who were categorised as having LTBI, any indications for LTBI screening (initiating or on immunosuppressive treatment, pre-organ transplantation, end-stage renal disease, silicosis, pulmonary fibrosis, HIV, people from high TB incidence countries, people who use drugs, homeless people, people in prisons or with a history of incarceration, and health care workers) and contraindications to LTBI treatment (hepatotoxicity risk (e.g., alcohol dependence or viral hepatitis) or history of severe allergic reaction to isoniazid and rifampicin) were recorded. Additionally, if the patient was offered, accepted, initiated, and completed LTBI treatment, and if they developed active TB, they were recorded. For patients who did not complete LTBI treatment, whether they were a national of a low-incidence (fewer than 10 cases per 100,000), intermediate-incidence (between 10 and 30 cases per 100,000) or high-incidence (over 30 cases per 100,000) country according to the WHO, data were collected [20].

### 2.4. Data Analysis

Data analysis was performed by J. O’Connell using Stata 16.0 (StataCorp. 2015) [21]. Descriptive analysis of the patient’s age, sex, nationality, the indication for screening and the outcomes of those with a positive IGRA was performed. Where the outcome of testing was unknown, these patients were excluded from calculations of the proportions of patients with active TB or LTBI. The prevalence of LTBI in Irish and non-Irish patients and patients grouped according to the indication for IGRA testing was calculated using the number of patients categorised with LTBI as the numerator and the number of patients without active TB or a history of treated TB infection as the denominator. Binary logistic regression was performed to determine if nationality, sex, or age were predictors of having TB infection. Binary logistic regression was performed to determine if there was an association between treatment non-completion and nationality, sex, or age. A chi-squared test was used to test for associations between categorical variables, except if the numerator or denominator of a categorical variable was less than five, in which instance, a Fisher’s exact test was used. A *p*-value of less than 0.05 was chosen to indicate statistical significance. 

## 3. Results

### 3.1. Description of Cohort Screened

There were 1681 IGRA tests performed in 1507 patients between 2016 and 2018. Half of those tested (754/1507) were men. The median age of patients at the time their first test was requested was 45 years (interquartile range (IQR) 33–58). Of all patients, 89.1% (1343/1507) were Irish and 10.9% (164/1507) were non-Irish. Overall, 4.8% (73/1507) of patients screened had a positive test (Table 1). There were 36 patients, all of whom had a negative IGRA, where the outcome of screening could not be extracted from their health care record due to insufficient information. 

Few patients (1.6% (23/1471) were categorised as having had active TB, 0.5% (7/1471) of patients had a history of previously treated TB infection, and 3.9% (58/1471) of patients, all of whom had a positive IGRA, were categorised as having had LTBI. The prevalence of LTBI was 4% (58/1441, 95% confidence interval (CI) 3.1–5.2%).

Of those screened for LTBI, patients on immunosuppressive treatment were the largest cohort (95.8% (1164/1215) of patients). Active TB was diagnosed in less than 1% (7/1164) of this cohort. Patients who had an IGRA performed in the context of an investigation for active TB were the second largest cohort. In these patients, a diagnosis of active TB was the outcome of testing in 5.9% (16/270).

Among Irish nationals, 83.4% (1251/1500) of tests were performed to screen for LTBI, 15.1% (226/1500) were performed during an investigation for active TB and 1.5% (23/1500) had an unknown indication. Among non-Irish nationals, 61.9% (112/181) of tests were performed to screen for LTBI, 30.4% (55/181) were performed during an investigation for active TB and 7.7% (14/181) had an unknown indication for testing. Those who were non-Irish nationals were more likely than those who were Irish nationals to have an IGRA performed while being investigated for active TB (30.4% (55/181) vs. 15.1% (226/1500), (χ^2^ (1, *n* = 1681) = 27.22, *p* < 0.001)).

Patients who were non-Irish were more likely to have a diagnosis of active TB compared with patients who were Irish (6.7% (10/149) vs. 1% (13/1318), odds ratio (OR) 7.2, 95% CI 3.1–16.8, *p <* 0.001). The prevalence of LTBI in the non-Irish cohort was 8.7% (12/138, 95% CI 4.6–14.7%), compared with a prevalence of 3.5% (46/1303, 95% CI 2.6–4.7%) in the Irish cohort. Patients who were non-Irish were more likely to have LTBI than patients who were Irish (8.7% vs. 3.5%, OR 2.6, 95% CI 1.3–5.0, *p* < 0.001). When considering patients screened prior to immunosuppressive treatments, which excludes patients investigated for active TB, there was no evidence of an association between LTBI and non-Irish nationality (3.1% (33/1074) vs. 3.8% (3/80) OR 1.23, 95% CI 0.4–4.1, *p* = 0.737). 

Among Irish nationals, male sex was a predictor of having LTBI (5.1% vs. 2.0%, OR 2.6, 95% 1.36–5.02, *p* < 0.005). However, among non-Irish nationals, there was no evidence that male sex predicted having LTBI (10% vs. 7.4%, OR 1.4, 95% 0.42–4.65, *p* = 0.58). Increasing age predicted having LTBI among both Irish nationals (OR 1.04, 95% CI 1.02–1.06, *p* < 0.001) and non-Irish nationals (OR 1.09, 95% CI 1.03–1.15, *p* < 0.05). 

### 3.2. Screening Effectiveness

Screening completion occurred in 97% (130/134) of patients in the randomly selected sample and, within this sample, screening completion occurred in 96.6% (114/118) of patients on immunosuppressive treatment, all patients being investigated for active TB (*n* = 13), both patients that were TB case contacts (*n* = 2) and the one patient who had radiological findings suggestive of TB infection. Four patients (3%) screened because they were on immunosuppressive treatment did not have chest radiographies performed, and as a result, were not deemed to have completed TB screening. All four patients had a negative IGRA, and no symptoms of TB had been documented. All patients in the random sample (*n* = 134) with symptoms of TB (*n* = 14) or an abnormal chest radiograph (*n* = 13) completed screening. In the entire cohort, all 73 patients with a positive IGRA completed screening. 

Overall, 40 patients had LTBI, an indication for LTBI treatment with no contra-indication to treatment. Of these, 67.5% (27/40) completed treatment for LTBI. Twenty percent (8/40) of patients eligible for treatment were not offered it by the health care provider, 2.5% (1/40) did not accept treatment when offered, and 10% (4/40) did not complete treatment after initiating it. 

The number of people screened for LTBI was 1215 (Figure 2), of whom 3.1% (38/1215) were diagnosed with LTBI. Of the patients who were not offered, did not accept, or did not complete LTBI treatment (18% (7/38)), none had their immunosuppressant treatment escalated to anti-TNF alpha treatment as planned. None of these patients developed active TB, with a mean follow-up after testing of 1.9 years (standard deviation (SD) = 0.9).

Evaluation of the cascade of IGRAs performed during investigations for active TB (Figure 3) identified six patients who did not have active TB, had an indication for LTBI treatment, and no treatment contra-indication. Of these six patients, three were men and three were women. The patient’s ages ranged from 30 to 53 years. All six of these patients were non-Irish and from countries with an intermediate or high TB incidence. 

### 3.3. Screening Efficiency

The cost per case of LTBI identified (*n* = 58) was EUR 2048, and the cost of screening per case who completed treatment (*n* = 27) was EUR 4400. The cost of identifying an Irish national with LTBI (*n* = 46) was EUR 2304 and the cost of screening per case who completed treatment (*n* = 23) was EUR 4576. For non-Irish nationals with LTBI (*n* = 12), the cost of identifying a case was EUR 1066 and the cost of screening per case who completed treatment (*n* = 4) was EUR 3322. 

### 3.4. Equity of Care

There were 27 and 13 patients eligible for LTBI treatment who were and were not treated for LTBI, respectively. The mean age of patients who were not treated for LTBI (*n* = 13) was 53.5 years (standard deviation (SD) 12.1), compared with an age of 55.7 years (SD 14.8) for patients who were treated for LTBI (*n* = 27), but there was no evidence of an association between age and LTBI non-treatment (Table 2). There were 26% (6/23) of men and 41% (7/17) of women not treated for LTBI. There was no evidence of an association between sex and LTBI non-treatment. Being a non-Irish national was a predictor for LTBI non-treatment (OR 6.86, 95% CI 1.35–34.70, *p* < 0.05).

## 4. Discussion

In this study, the quality of LTBI screening in a tertiary centre in Ireland, which primarily included patients on immunosuppressive treatments, was evaluated in terms of effectiveness, efficiency, and equity. Regarding effectiveness, screening completion was high (97%) and similar to that reported in the literature for patients screened for medical reasons (98%) [14]. However, most patients intended for LTBI screening are lost prior to screening initiation [14], which could not be measured in this study. The proportion of patients who started but did not complete LTBI treatment in this study (13%) was comparable with that reported in the literature for patients on immunosuppressive treatments (0–15%) [22,23,24,25]. The proportion of patients not initiated on LTBI treatment (22.5%) is higher than that reported in the literature for patients screened for medical reasons (15%) [14]. Patients tested during an investigation for active TB and found to have LTBI were not offered treatment, all of whom were non-Irish nationals. Being a non-Irish national with LTBI was a predictor of not being treated, demonstrating that the care provided was inequitable. A survey of health care providers in this centre found that they reported a low perception of the risk of TB, low confidence in performing aspects of LTBI treatment, and used multiple different guidelines or no guidelines at all for LTBI care [26]. In Ireland, national clinical guidelines are over a decade old and there is no education module on TB for health care providers, despite the need previously being identified [27,28,29,30]. In other low-incidence settings, low health care provider knowledge about the need for LTBI treatment has frequently been reported as a barrier to LTBI treatment [14]. In Europe, variable knowledge or acceptance of guidelines has also been reported as a barrier to LTBI treatment, with the engagement of expert opinion leaders and continuous education of health care providers being possible solutions [31]. In England, training and education on LTBI screening and treatment were reported to facilitate health care providers delivering care as part of the national LTBI programme [32].

The measures of efficiency demonstrated the substantial cost of identifying patients with LTBI in this low-prevalence cohort, reinforcing the importance of maximising progression to LTBI treatment completion, particularly if LTBI screening were to be scaled and expanded in this tertiary centre as part of a national programme for LTBI management. The absolute number of patients tested using an IGRA (*n* = 1507), screened for LTBI using an IGRA (*n* = 1215), and subsequently treated for LTBI to completion (*n* = 27) over a three-year period was small, given that the population of the tertiary hospital catchment area is over 287,000 [33]. The prevalence of LTBI among the entire cohort was only 4%, consistent with that reported among patients screened due to immunosuppressive treatment use in other centres in Ireland [11]. Individually, the patients treated for LTBI had their risk of TB reduced because it is known to be highly effective [7,8]. However, to achieve population-level benefits in TB control with the aim of eliminating TB, this tertiary centre should performed LTBI screening and treatment at scales programmatically among high-risk population groups [6], as occurs in England where TB clinics are engaged in programmatic LTBI management in people from countries with a high incidence of TB (≥150/100,000; or sub-Saharan Africa) [15]. Other indications for LTBI treatment according to the Guidelines for the Prevention and Control of Tuberculosis 2010 [34], such as being from a country with a high incidence of TB, being homeless, or being a person who uses intravenous drugs were infrequent or did not feature at all among the indications for screening in this study, suggesting that these cohorts are not being screened for LTBI in this tertiary centre. Although there are few prevalence studies among such risk groups in Ireland [11], in England, IGRA positivity during programmatic screening was observed in 32% of immigrants from high-incidence countries, a group that contributed disproportionately to the national TB burden [14]. The TB service in this tertiary centre should expand screening to groups with a higher prevalence of LTBI, but it is unclear which population groups, and national guidance is lacking in this area. Additionally, programmatic LTBI screening and treatment based on risk related to population characteristics (e.g., country of origin), rather than individual characteristics (e.g., being on an immunosuppressant), may require greater prioritisation of the value of the population benefit of LTBI treatment by health care providers when making clinical decisions [31]. National clinical guidelines for LTBI screening and treatment should reflect this.

### Strengths and Limitations

The strengths of this study are that it included patients screened over three years. Although the sample size was small, it is larger than that of most previous similar studies conducted in Ireland [11]. This study has several limitations. Regarding internal validity, most of the data were collected retrospectively from health care records that could incompletely or inaccurately reflect the LTBI screening and treatment performed. This risk of measurement bias was reduced but not eliminated by cross-checking data collected across multiple components of the patient’s health care record (e.g., patient file and clinic letters). Measurement bias due to variability between data collectors when interpreting the health care records cannot be excluded. Standard definitions of outcomes, and the cross-checking of all data collected for accuracy by one data collector, may have reduced the risk of this bias. The generalisability of this study is limited because it was performed in a single centre in tertiary care, and so may have had a select sample of patients different from other centres, with screening and treatment practices different from those in other centres. The dataset did not include health care workers screened for LTBI in the occupational health department, who are an important risk group in which to identify LTBI. This study was observational and cannot infer causation. However, the literature to support potential explanations of the observations made has been discussed. A weakness of this research was that it measured the quality of LTBI screening according to only three of the seven domains in the adapted WHO version of the Institute of Medicine framework. LTBI screening in this tertiary centre may be of low quality in other regards (e.g., by measures of patient-centredness or integration,) but would not have been detected by this study. Qualitative research could be performed to evaluate the patient-centredness of LTBI screening in this centre.

## 5. Conclusions

This study evaluated the quality of LTBI screening using a framework, and identified that LTBI screening in this tertiary centre needs to be scaled and expanded and that treatment initiation needs to be improved, particularly among non-Irish nationals. 

## Figures and Tables

**Figure 1 tropicalmed-07-00019-f001:**
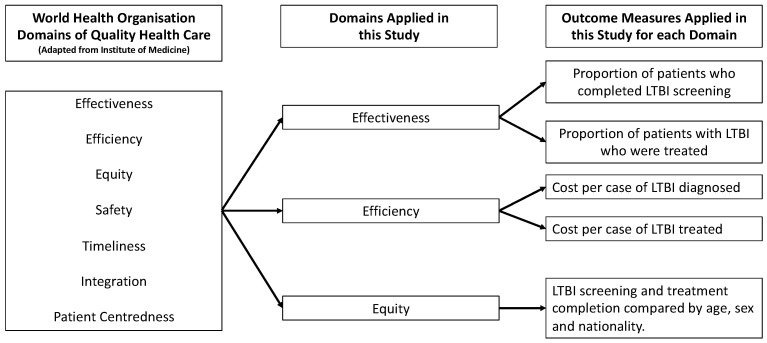
Framework, domains, and measures applied.

**Figure 2 tropicalmed-07-00019-f002:**
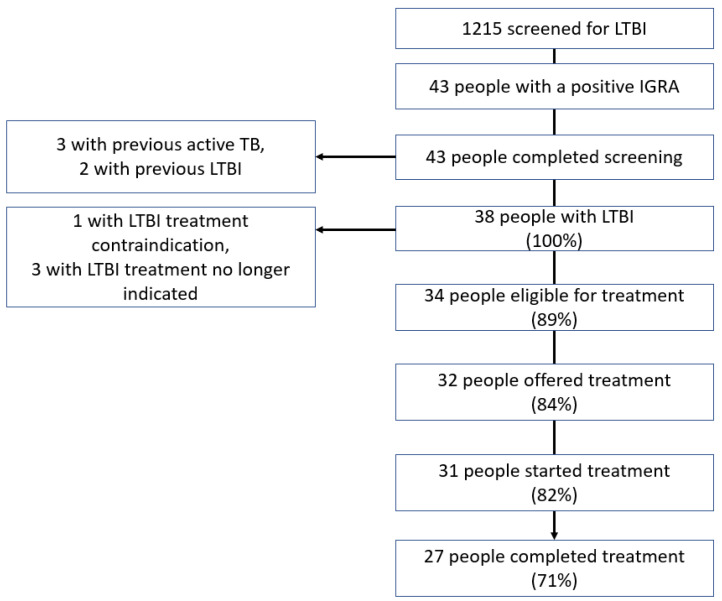
Cascade of interferon-gamma release assay testing performed to screen for latent tuberculosis infection.

**Figure 3 tropicalmed-07-00019-f003:**
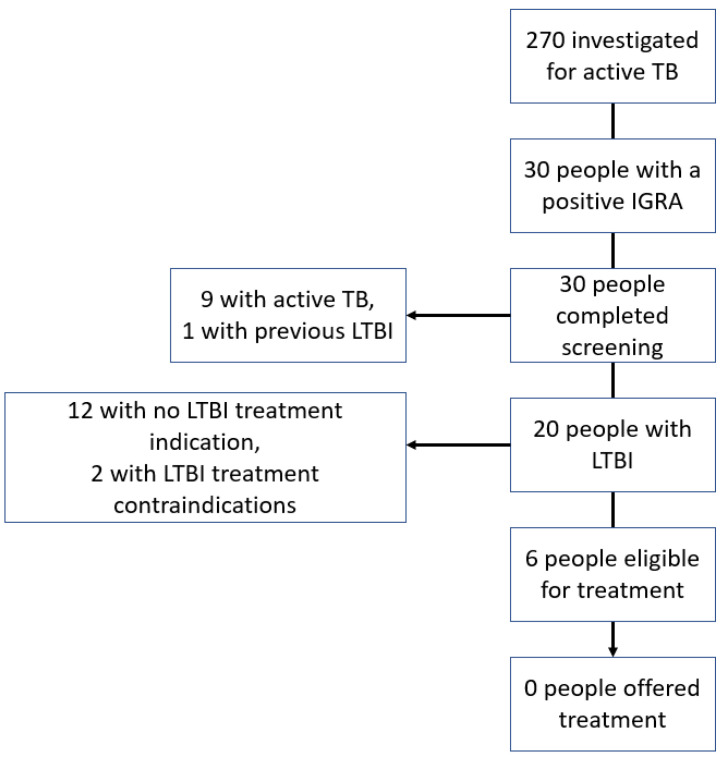
Cascade of interferon-gamma release assay testing performed during investigation for active tuberculosis.

**Table 1 tropicalmed-07-00019-t001:** Indications for screening and outcomes of IGRA testing.

Indication Group	Number of Tests (Proportion of Total Tests)	Positive IGRA (Proportion of Tests in Indication Group)	Number of Individuals (Proportion of Total Individuals)	Individuals Testing Positive (Proportion of Individuals in Indication Group)
Immunosuppression therapy	1312 (78%)	44 (3.3%)	1164 (77.2%)	40 (2.7%)
Investigation for active TB	281 (16.7%)	32 (11.4%)	270 (17.9%)	30 (11.4%)
Recent TB case contact	28 (1.7%)	2 (7.1%)	25 (1.7%)	2 (8%)
HIV	9 (<1%)	0	9 (<1%)	0
Pre-organ transplantation	9 (<1%)	1 (11.1%)	8 (<1%)	1 (12.5%)
Person from high incidence TB country	2 (<1%)	0	2 (<1%)	0
Radiological finding of LTBI	3 (<1%)	0	3 (<1%)	0
Unknown	37 (2.2%)	0	36 (2.4%)	0
Total	1681	79	1507 *	73

* Sum of column values does not equal 1507 because some patients had more than one test for more than one indication.

**Table 2 tropicalmed-07-00019-t002:** Predictors of not being treated for latent tuberculosis infection.

Variable	Not Treated for LTBI (*n* = 13)	Treated for LBTI (*n* = 27)	Odds Ratio (Not Treated for LTBI vs. Treated for LTBI)
Age (years, mean ± SD)	53.5 ± 12.1	55.7 ± 14.8	0.98 (95% CI 0.94–1.03, *p* = 0.64)
Sex			
Men	26% (6/23)	74% (17/23)	0.50 (95% CI 0.13–1.92, *p* = 0.317)
Women	41% (7/17)	59% (10/17)	
Nationality			
Non-Irish	67% (6/9)	33% (3/9)	6.86 (95% CI 1.35–34.70, *p* < 0.05)
Irish	23% (7/31)	77% (24/31)	

## Data Availability

Data were collected as part of clinical audit and are not available for dissemination.

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
