# Peer review of "Evaluating the Quality of Latent Tuberculosis Infection Screening in Ireland: A Single-Centre Retrospective Cohort Study"

_tropicalmed, 2022, doi:10.3390/tropicalmed7020019_

Round 1
Reviewer 1 Report
Thank you for the opportunity to review the manuscript, An evaluation of the quality of latent tuberculosis infection screening and treatment in a tertiary centre in Ireland, submitted to Tropical Medicine and Infectious Disease. Overall, I believe this is a useful descriptive study with an evaluation of screening and LTBI treatment. There are a number of revisions required prior to publication.
For the revision process, I request the authors provide the completed STROBE checklist for cross-sectional studies as many of the limitations I identified in the manuscript are included in this reporting guideline (https://www.equator-network.org/reporting-guidelines/strobe/). In the section below, I am providing specific areas with important limitations that need to be addressed. Addressing these areas will improve the presentation and overall quality of the reporting elements of the study. I look forward to reviewing the revised manuscript.
INTRODUCTION
The introduction provides no information about the context of the study. First, most readers of this journal are not familiar with the TB context in Ireland, or the larger context of the surrounding European Union. As such, I recommend one to two paragraphs providing this information. Second, there is no scientific justification for the study. For this reason, I recommend providing a paragraph with the scientific rationale for this study. Third, the area specific to the model guiding the research is not clearly defined. The reference (4) is not specific to the WHO as noted in the manuscript. Instead, this is a seminal work from the United States. This information needs to be corrected as well as clearly and concisely explained in the context of the previous information in the introduction. IMPORTANTLY, I recommend the model be labeled as a subsection in the introduction. For the methods section, a figure with the model and the associated variables from this study could be useful as the methods are not clear.
Throughout the manuscript, the use of quality is not consistent and clear. When addressing screening, the most commonly used term is screening adequacy. Similarly, the term treatment is fine but connecting treatment to quality is not advisable as the quality of treatment is not measured by this study. In the next area, there are a number of statements from the manuscript to demonstrate the lack of precise language. There area multiple additional examples of inconsistencies due to the lack of precise language that need to be corrected through the manuscript.
Title - An evaluation of the quality of latent tuberculosis infection screening and treatment in a tertiary centre in Ireland.
Line 175 - In this study, the quality of LTBI care in a tertiary centre in Ireland was evaluated in 175 terms of effectiveness, efficiency, and equity.
Line 228 - "This local quality of care evaluation adds to the research nationally in this area."
Line XXX - "The quality of LTBI screening in a tertiary centre in Ireland has been measured according to the domains of effectiveness, efficiency, and equity."
METHODS
This section is the weakest of the manuscript due to the lack of minimum reporting criteria. The methods section would be strengthened by reorganizing the information into subsections, research design, setting and sample, data collection, and data analysis. The STROBE checklist will assist the authors in making certain all the reporting elements are provided in the revised manuscript.
DISCUSSION
The first paragraph of the discussion contains too much repetitive information from the results which needs to be removed.
There are multiple statements about the findings from the results without reference to other published literature. For example, the statement (line 197) "The prevalence of LTBI was 197 greater among non-Irish nationals than Irish nationals in the entire cohort, but only when 198 patients with other risk factors for TB infection, not measured in this study, were included." needs to have citations from the literature from Ireland as well as other comparative countries. This paragraph provides no citations specific to this important finding.
There only 15 references for this manuscript, with very few comparative citations from other countries with comparable contexts. As I work in the area of TB, I am familiar with the international literature. Please strengthen the discussion with the comparable country findings specific to this study. As the authors previously completed a systematic review about LTBI, I am sure they are familiar with the international context specific to my recommendation.
LIMITATIONS
The limitations of the study were addressed. I recommend this be labeled as a subsection (Strengths and Limitations) at the end of the discussion. But, the statement (line 217), "The strengths of this study are that it included a large sample size and included patients screened over three years." is not correct as the LTBI sample was not large. Also, there were many biases specific to observational cross-sectional designs not addressed. As such, I recommend the authors review the biases commonly associated with this type of work to provide the additional disclosure.
LAST PARAGRAPH (following Strengths and Weaknesses)
The last paragraph (line 228), beginning with "This local quality of care evaluation adds to the research nationally in this area." is not discussion. This information may be applicable to the conclusion.
CONCLUSION
The conclusion is not impactful nor useful for understanding the key findings from the study. The last paragraph of the discussion might be this information as there are policy implications.
Author Response
Department of International Health and Tropical Medicine,
Royal College of Surgeons in Ireland,
Dublin 2,
Dublin,
Ireland
Date typed: January 21st 2022
Dear Editor,
Thank you for considering our manuscript. We found the reviewers comments constructive and have been able to improve the manuscript significantly having considered them. Please find the revised manuscript which we hope is acceptable for publication and below our response to the peer-reviewer comments.
Reviewer 1.
For the revision process, I request the authors provide the completed STROBE checklist for cross-sectional studies as many of the limitations I identified in the manuscript are included in this reporting guideline (https://www.equator-network.org/reporting-guidelines/strobe/).
We have revised the manuscript so that it reports all items in the STROBE checklist for cohort studies (this study did include information on subjects at more than one point in time, so strictly speaking it was not cross-sectional but we agree the use of a STROBE checklist was beneficial and we have included this as an appendix and referenced it in the methods section).
The introduction provides no information about the context of the study. First, most readers of this journal are not familiar with the TB context in Ireland, or the larger context of the surrounding European Union. As such, I recommend one to two paragraphs providing this information.
We have included a paragraph to describe the importance of latent tuberculosis screening and treatment for TB elimination in low-incidence countries (paragraph 1, line 38). We have included a paragraph that describes the basic TB epidemiology and current approach to latent TB care in Ireland (paragraph 2).
Second, there is no scientific justification for the study. For this reason, I recommend providing a paragraph with the scientific rationale for this study.
The scientific rationale behind performing evaluations of latent TB screening and treatment is described in paragraph 3.
Third, the area specific to the model guiding the research is not clearly defined. The reference (4) is not specific to the WHO as noted in the manuscript. Instead, this is a seminal work from the United States. This information needs to be corrected as well as clearly and concisely explained in the context of the previous information in the introduction. IMPORTANTLY, I recommend the model be labeled as a subsection in the introduction.
A separate subsection to describe the model has been included in the introduction, paragraph 4.We have explained in this section that the model is that of the Institute of Medicine but it has been utilized and adapted by the World Health Organization who added a 7th domain (integration). The references are now correct.
For the methods section, a figure with the model and the associated variables from this study could be useful as the methods are not clear.
Figure 1 has been added to the methods section to describe how the model, the domains chosen and the measures applied in this study are related.
Throughout the manuscript, the use of quality is not consistent and clear. When addressing screening, the most commonly used term is screening adequacy.
We acknowledge that screening adequacy is a term applied in other contexts. We prefer the term quality, because it encompasses other aspects of screening such as cost and equity, which aren’t captured by the term adequacy, consistent with the Institute of Medicine framework applied in the study. As such, we would prefer to retain this terminology.
Similarly, the term treatment is fine but connecting treatment to quality is not advisable as the quality of treatment is not measured by this study.
We did not measure all components of the quality of treatment, but whether or not the patient completed treatment is, in our opinion, a basic measure of the effectiveness of care provided. It is necessary to report the proportion of patients completing treatment when discussing the effectiveness of screening because all steps are interlinked i.e, screening is only effective if it identifies patients with latent TB who go on to complete treatment. Nonetheless, we have edited the manuscript to only use the term quality with reference to screening effectiveness, efficiency and equity and not with regard to latent TB treatment.
In the next area, there are a number of statements from the manuscript to demonstrate the lack of precise language. There area multiple additional examples of inconsistencies due to the lack of precise language that need to be corrected through the manuscript.
We have revised the manuscript to make the language more concise and consistent throughout, including the three specific examples described by the reviewer (see below)
Title - An evaluation of the quality of latent tuberculosis infection screening and treatment in a tertiary centre in Ireland.
Changed to:
Evaluating the Quality of Latent Tuberculosis Infection Screening in Ireland: A Single-centre Retrospective Cohort Study
Line 228 – “This local quality of care evaluation adds to the research nationally in this area.”
Line removed.
Line XXX - "The quality of LTBI screening in a tertiary centre in Ireland has been measured according to the domains of effectiveness, efficiency, and equity."
Changed to:
In this study, the quality of LTBI screening in a tertiary centre in Ireland, which primarily included patients on immunosuppressive treatments, was evaluated in terms of effectiveness, efficiency, and equity.
This section is the weakest of the manuscript due to the lack of minimum reporting criteria. The methods section would be strengthened by reorganizing the information into subsections, research design, setting and sample, data collection, and data analysis. The STROBE checklist will assist the authors in making certain all the reporting elements are provided in the revised manuscript.
We have expanded the methods section significantly and used the subsection headings as suggested by the reviewer and consistent with the STROBE checklist.
The first paragraph of the discussion contains too much repetitive information from the results which needs to be removed.
We have removed this repetitive information. The discussion now focuses on two findings from the study- the need to improve provider recommendation of treatment and the need to scale and expand LTBI screening at this centre.
There are multiple statements about the findings from the results without reference to other published literature. For example, the statement (line 197) "The prevalence of LTBI was 197 greater among non-Irish nationals than Irish nationals in the entire cohort, but only when 198 patients with other risk factors for TB infection, not measured in this study, were included." needs to have citations from the literature from Ireland as well as other comparative countries.
This paragraph has been removed.
There only 15 references for this manuscript, with very few comparative citations from other countries with comparable contexts. As I work in the area of TB, I am familiar with the international literature. Please strengthen the discussion with the comparable country findings specific to this study.
We have increased the number of citations to 35, referencing studies from other low incidence settings when discussing the two most important findings mentioned in the first two paragraphs of the discussion.
The limitations of the study were addressed. I recommend this be labeled as a subsection (Strengths and Limitations) at the end of the discussion.
We have done this.
But, the statement (line 217), "The strengths of this study are that it included a large sample size and included patients screened over three years." is not correct as the LTBI sample was not large.
We have corrected this by saying instead “The strengths of this study are that it included patients screened over three years. Although the sample size was small, it is larger than that of most previous similar studies conducted in Ireland”
Also, there were many biases specific to observational cross-sectional designs not addressed. As such, I recommend the authors review the biases commonly associated with this type of work to provide the additional disclosure.
We have expanded the limitations section to discuss the limitations of our study more.
The last paragraph (line 228), beginning with "This local quality of care evaluation adds to the research nationally in this area." is not discussion. This information may be applicable to the conclusion.
This paragraph has been deleted.
The conclusion is not impactful nor useful for understanding the key findings from the study. The last paragraph of the discussion might be this information as there are policy implications.
The conclusion has been strengthened to reflect the initial study aim and the implications of the findings. It now reads:
This study evaluated the quality of LTBI screening using a framework and identified that LTBI screening in this tertiary centre needs to be scaled and expanded and that treatment initiation needs to be improved, particularly among non-Irish nationals.
We hope these changes are satisfactory.
Yours Sincerely,
James O’Connell
Reviewer 2 Report
The paper sought to assess the quality of screening and treatment for latent TB infection at a tertiary hospital in Ireland. This was a retrospective review and found that the initiation of treatment was not initiated for those with latent TB infection.
comments:
The first section of results is unclear :what doe " half of these tested"refer to?
the sentence referring to Table 1 is unclear.
The conclusion in the abstract and the paper should reflect the aim. The current conclusion is made up of the aim and the recommendation.
Author Response
Department of International Health and Tropical Medicine,
Royal College of Surgeons in Ireland,
Dublin 2,
Dublin,
Ireland
Date typed: January 21st 2022
Dear Editor,
Thank you for considering our manuscript. We found the reviewers comments constructive and have been able to improve the manuscript significantly having considered them. Please find the revised manuscript which we hope is acceptable for publication and below our response to the peer-reviewer comments.
REVIEWER 2
The first section of results is unclear :what doe " half of these tested"refer to?
It should have read that “half of those tested were men”. We have corrected this.
the sentence referring to Table 1 is unclear.
We have re-written this sentence so that it now reads more clearly as
Overall, 4.8% (73/1507) of patients screened had a positive test (Table 1).
The conclusion in the abstract and the paper should reflect the aim. The current conclusion is made up of the aim and the recommendation.
The conclusion now reflects the aim more clearly. It summarizes the two main implications of the findings from the study.
We hope these changes are satisfactory.
Yours Sincerely,
James O’Connell

Round 2
Reviewer 1 Report
​ After carefully reviewing the revisions, the manuscript is suitable for publication. Overall, the authors did a good job addressing the recommendations. ​Although the data was collected over a two-year period, there was not a longitudinal comparison of specific points of data hence the reason the observational study design was most compatible with the Strobe for cross-sectional studies. Importantly, the resulting methods section provides a better description about what was done and why in a structured manner. The revisions made the manuscript clearer and more concise in presenting an interesting study with relevance to an international audience. The new figure is quite helpful to understand the relationship of the conceptual model with the research question resulting in the operationalization of the variables. The discussion section provides additional insights for the international audience in terms of the broader literature. Finally, the limitations are more specifically addressed to close the discussion. Please review the text as there seem to be a couple of spacing and punctuation issues due to the track changes. Again, good work with the revisions.